# Dynamic Top-k Estimation Consolidates Disagreement between Feature Attribution Methods

**Jonathan Kamp**[1]     **Lisa Beinborn**[1]     **Antske Fokkens**[1,2]

[1]Computational Linguistics and Text Mining Lab, Vrije Universiteit Amsterdam
[2]Dept. of Mathematics and Computer Science, Eindhoven University of Technology
`{j.b.kamp,l.beinborn,antske.fokkens}@vu.nl`

## Abstract

Feature attribution scores are used for explaining the prediction of a text classifier to users by highlighting a $k$ number of tokens. In this work, we propose a way to determine the number of optimal $k$ tokens that should be displayed from sequential properties of the attribution scores. Our approach is dynamic across sentences, method-agnostic, and deals with sentence length bias. We compare agreement between multiple methods and humans on an NLI task, using fixed $k$ and dynamic $k$. We find that perturbation-based methods and Vanilla Gradient exhibit highest agreement on most method–method and method–human agreement metrics with a static $k$. Their advantage over other methods disappears with dynamic $k$s which mainly improve Integrated Gradient and GradientXInput. To our knowledge, this is the first evidence that sequential properties of attribution scores are informative for consolidating attribution signals for human interpretation.

## 1 Introduction

Feature attribution scores are a glimpse behind the scenes of neural models, or at least that is the promise. Various interpretability methods have been developed that can generate attribution scores to interpret the degree to which a language model's features (tokens) contributed to the predicted label. However, attribution values from different methods can vary considerably even on the same instance (Madsen et al., 2022, i.a.). Contradictory interpretations cast doubts on their usefulness and reliability in a practical setting.

When comparing attribution methods, the focus commonly lies on assessing agreement with human explanations (often referred to as plausibility (Jacovi and Goldberg, 2020)) and on agreement between methods (Neely et al., 2022; Krishna et al., 2022). However, the evaluation procedures have not been standardised yet and varying influential

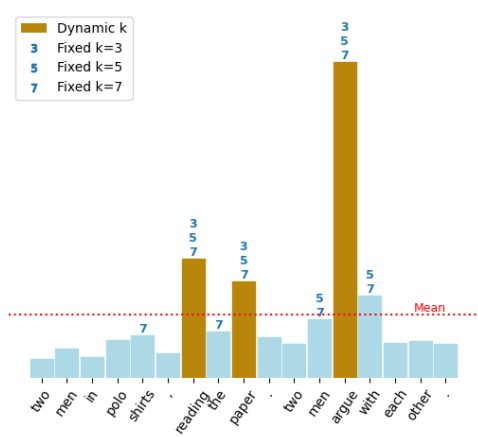

Figure 1: Dynamic $k$ versus fixed $k$ (Vanilla Gradient).

factors such as the exact task, data, and model have led to contradictory conclusions. For example, Attanasio et al. (2022) find that perturbation-based methods better agree with human preferences than gradient-based methods, while Atanasova et al. (2020) report the reverse tendency.

A factor that may also influence results but has been largely understudied is the impact of the chosen number of $k$ most salient tokens that are taken into consideration. Previous analyses either use a fixed value of $k$ or determine the most salient tokens based on a fixed threshold for the attribution values. A clear drawback of using a fixed number of $k$ is that it can result in excluding tokens with scores close to the top-$k$ and including tokens with low scores, disregarding significant score gaps with truly important tokens. Jesus et al. (2021) set $k$ to a fixed value of six and examine the effect of visualizing the most salient features on decision-making accuracy, time, and agreement for a fraud analysis task. Bastings et al. (2022) set $k$ to the low fixed values of 1 and 2 as their analysis focuses on the identification of specific shortcut cues in a closed experimental setup. Camburu et al. (2019) determine $k$ using a fixed threshold and identify

tokens as salient if their attribution is $> 0.1$. When visualizing attributions to users, they use constant values of $k$ (5 and 10). Absolute attribution scores tend to decrease for sentences with a larger number of tokens which is not captured by their threshold-based approach. Krishna et al. (2022) determine $k$ as a $1/4$ fraction of the average sentence length in the data and set it to 11.

As token-level agreement generally increases the closer $k$ gets to the number of tokens in a sentence (see Appendix C for a visualisation of the *sentence length bias*), the evaluation of attribution methods is clearly top-$k$ dependent. Li et al. (2020) indicate that intrinsic model evaluation is generally sensitive to different values of $k$ but do not measure its effect on agreement. Both Neely et al. (2022) and Krishna et al. (2022) find consistent disagreement between attribution methods, but do not account for the influence of $k$.[1]

**Contributions** In this paper, we systematically explore the role of $k$ on the observed method–method agreement and method–human agreement for extracting explanations for a natural language inference task. For this purpose, we develop the new metric agreement@$k$. We propose to determine $k$ dynamically for each method and each instance based on the sequential properties of the attribution profile. Our approach is inspired by methods for event detection and detects attribution peaks in a method-agnostic fashion that circumvents the sentence-length bias by allowing different values of $k$ for each instance. We find that:

- agreement at the token level is sensitive to different values of $k$ and the effect varies across attribution methods;

- determining $k$ with respect to attribution profiles consolidates the disagreement between attribution methods, in particular for Integrated Gradient and GradientXInput.

We take a novel perspective on human attribution evaluation by interpreting it as a ranking task.

## 2 Experimental Setup

We fine-tune a model on a natural language inference task and analyze the agreement between feature attribution methods.

**Data** We use the e-SNLI dataset with the default split of 549,361 instances for training, 9,842 for development, and 9,824 for testing (Camburu et al., 2018). Each instance consists of a premise, a hypothesis, and an output label that indicates the semantic relation between the premise and the hypothesis: contradiction, entailment, neutral. Each premise is paired with three hypotheses (one for each label) to obtain balanced classes. Instances span 21 tokens on average (5–113). 6,325 annotators highlighted tokens that they found most important to explain the gold label (avg. number of highlighted tokens: 4±3). We used the dev and test instances which were annotated by at least three annotators (we used dev for exploration, and test for our experiments).

**Backbone Model** We fine-tune DistilBERT (Sanh et al., 2019) using ten different random seeds and select the median model, i.e., the model with the least variation in attribution profiles compared to the other nine models, for further analysis.[2] The model yields a performance of 0.89 F1 on both the dev and test set.

**Feature Attribution** We use the Ferret package v0.4.1 (Attanasio et al., 2023) to calculate attributions using the gradient-based methods Vanilla Gradient (Simonyan et al., 2014) and Integrated Gradient (Sundararajan et al., 2017), and the perturbation-based methods Partition SHAP (Lundberg and Lee, 2017) and LIME (Ribeiro et al., 2016). For the gradient-based methods, we use both the plain gradients and the gradientXInput version for which the gradient is multiplied by the input token embeddings (Shrikumar et al., 2017). Hence, we examine a total of six methods.

**Evaluation** Recent studies evaluate feature attributions as a ranking task. Atanasova et al. (2020) calculate the mean average precision (MAP) compared to human labels and Bastings et al. (2022) restrict the evaluation to the top $k$ ranks (MAP@k) but it remains an open question how $k$ is selected.

An attribution method $\mathbf{A}$ assigns an attribution vector $\mathbf{a} = \{a_1, a_2, ..., a_n\}$ to a sentence consisting of tokens $\mathbf{s} = \{w_1, w_2, ..., w_n\}$ so that each $a_i$ indicates the salience of token $w_i$ for the predicted output label. We determine the $topk_A = \{t_1, t_2, ..., t_k\}$ by selecting the $k$ tokens with the highest attribution values. We propose a new metric

---

[1]All analyses are available at: https://github.com/jbkamp/repo-Dynamic-K.

[2]See Appendix A for implementation details on the model selection.

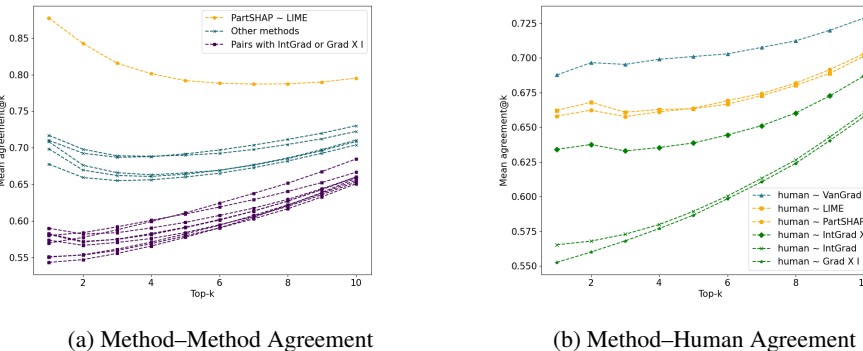

(a) Method–Method Agreement          (b) Method–Human Agreement

Figure 2: Mean agreement@$k$ for pairs of attribution methods (left) and between methods and humans (right).

for comparing $m$ attribution methods $A_1, ..., A_m$ by calculating sentence-level agreement@$k$ based on token relevance. Relevance for a token $w_i$ is determined by the ratio of methods that include the token in the *topk*. A high relevance score indicates that a token is assigned high attribution by many of the compared methods.

$$\text{Relevance } r(w_i) = \frac{\sum_{A_j=1}^{m} [w_i \in topk_{A_j}]}{m} \quad (1)$$

$$\text{Agreement@}k(s_i) = \frac{\sum_{w_i=1}^{n} r(w_i)}{\sum_{w_i=1}^{n} [r(w_i) > 0]} \quad (2)$$

This way, tokens not included in the top-$k$ selection of any method are assigned a relevance score of zero and are not considered in the calculation of agreement@$k$. This approach avoids artificially inflating the agreement@$k$ score by having high agreement on non-relevant tokens. The agreement@$k$ for a dataset of sentences $\mathbf{D} = \{s_1, s_2, \ldots, s_d\}$ is computed by averaging the sentence-level agreement@$k$ scores.

$$\text{Agreement@}k(D) = \frac{\sum_{s_i=1}^{d} \text{agreement@}k(s_i)}{d} \quad (3)$$

We evaluate our experiments comparing mean agreement@$k$ between the six attribution methods on the test data of e-SNLI.

## 3  Agreement at Fixed $k$

We compare attribution methods with each other and with human labels and analyze the role of $k$.

**Method–Method Agreement**   We identify three groups of mean agreement@$k$ across pairs of attribution methods in Figure 2a. The agreement between Partition SHAP–LIME clearly stands out (yellow line), in particular for small $k$. All comparisons involving Integrated Gradient or Gradient-XInput end up in the group with the least agreement (purple lines) and the remaining pairs obtain medium-level agreement (green lines). Agreement increases for bigger $k$ for pairs of methods with low to medium agreement. Our results contrast previous work which identified higher pairwise agreement between gradient-based methods compared to perturbation-based methods (Krishna et al., 2022).

**Method–Human Agreement**   We calculate token relevance for the three human annotators as the ratio of annotators who selected the token, therefore in the range $[0, .33, .67, 1]$. When we compare the attribution methods to human annotations, we find that the two perturbation-based methods lead to higher agreement than the gradient-based ones (Figure 2b) and that higher values of $k$ generally lead to better agreement. Our findings are partly in line with Attanasio et al. (2022) in that perturbation-based methods are more plausible than most gradient-based methods, and partly with Atanasova et al. (2020) for finding that Vanilla Gradient agrees more with human rationales than perturbation-based methods which in turn agree more with human than most of the other gradient-based methods. We contrast Ding and Koehn (2021) who find higher plausibility for Integrated Gradient over Vanilla Gradient. While consistency in performance across studies sheds light on the interrelatedness between methods, it is important to exercise caution when generalizing evaluation results across different models, datasets, and tasks.

## 4 Dynamic Top-$k$ Estimation

We have seen that the value of $k$ has a strong influence on the agreement between methods. Perturbation-based methods are more in line with human annotations for smaller settings of $k$ while the attribution profiles of gradient-based methods require relatively larger settings of $k$ to obtain a similar reflection of human preferences. We propose to determine the number of $k$ salient tokens dynamically based on the attribution profiles of the methods.

Inspired by event detection in time series (Taylor (2000), Palshikar et al. (2009), e.g.), we consider attribution profiles as sequences of token-level scores that indicate the local presence or absence of a peak. Each peak is a point in the sequence (i.e., the sentence) to which the model attributed a higher salience compared to neighboring points. We apply peak detection based on local maxima in the attribution profile to estimate a $k$ that is dynamic across method–instance combinations. The attribution profile of each individual method thus serves as the indicator of its peaks. A local maximum is defined as a point $x_i$ in a sequence such that it is greater than both its immediate left neighbor, $x_{i-1}$, and its immediate right neighbor, $x_{i+1}$. We additionally enforce the constraint that $x_i$ needs to be higher than the mean attribution of the sequence, corresponding to above-average model behavior. With our dynamic $k$ approach, we favor relative differences in attribution values over absolute thresholds for identifying the top-$k$ tokens. Figure 3 shows

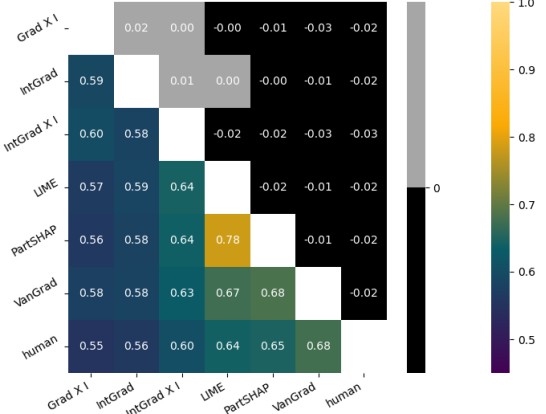

Figure 3: Mean agreement@$k$: dynamic $k$ (bottom left) and difference (top right) compared to fixed human average $k = 4$. The brighter the color, the higher the agreement.

that when determining $k$ dynamically, the agreement of Integrated Gradient and GradientXInput with the other methods increases. We find that dynamic $k$ indeed varies across instances and across methods. For example, attributions by Partition SHAP leads to fewer local maxima and therefore to lower values for dynamic $k$ (4.5±1.7 on average) compared to IntegratedGradient (7.3±2.6). We note that these ranges are close to human preferences for $k$ on the NLI task (4±3).

## 5 Further Insights

We briefly discuss complementary insights to the main results, as well as two examples of dynamic $k$ on instances from the dataset.

**Average Human Preference** Human annotations of token-level explanations were available for this task and we compared the effect of dynamic $k$ to the average number of annotations. However, these types of annotations are expensive, which often leads to selecting a fixed $k$. Normally, a valid approach is choosing a fixed $k$ that is close to the average number of annotated tokens per sentence, as the resulting ranges of dynamic $k$ for our methods reflected. In absence of annotations, this average number is unknown.

Dynamic $k$ highlights the implications of choosing a fixed $k$ that deviates from the average. We observe a clear distinction between lower and higher than average for GradientXInput and for Integrated Gradient. Table 1 reports the improvement on mean agreement@$k$ by the dynamic approach over fixed values. Generally, the lower fixed $k$, the more improvement we observe. For $k > 5$ there is no improvement.

| $k =$ | 1 | 2 | 3 | 4 | 5 |
|---|---|---|---|---|---|
| **Grad X I** | +.10 | +.08 | +.05 | +.02 | +.01 |
| **IntGrad** | +.01 | +.04 | +.02 | +.01 | +.00 |

Table 1: Absolute difference on mean agreement@$k$ by dynamic $k$ over fixed $k$ (GradientXInput and Integrated Gradient). Summed method–method agreement scores are reported. Dynamic $k$ is compared to different values of fixed $k \in [1, 2, 3, 4, 5]$.

This further suggests that, in the absence of human annotations, dynamic $k$ may provide an estimate of the average human preference for $k$.

**Peaks as Signals** We analyse two examples to better understand the dynamic $k$ approach. Figures

1 and 4 show the application of dynamic $k$ versus fixed $k$ set to 3, 5, and 7 for two different methods.

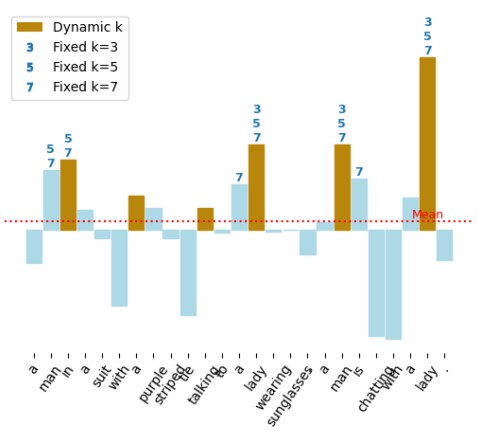

Figure 4: Dynamic $k$ versus fixed $k$ (Integrated Gradient).

In Figure 1, dynamic $k$ highlights the same top three tokens that have been selected by fixed $k$, namely *reading*, *paper* and *argue*. Tokens that are only selected by $k = 5$ and $k = 7$ are not included as they do not form peaks in the attribution profile.

Figure 4 illustrates differences in the attribution patterns captured by fixed $k$ and dynamic $k$. While fixed $k$ selects any tokens in descending order of attribution score, dynamic $k$ captures *signals*. It highlights two peaks that were not captured by fixed $k$ (due to their lower absolute values) because they stand out relative to the surrounding tokens. Signals can be described as tokens that represent a salient part or phrase in the sentence beyond the token level (e.g. *a man in a suit*; *a lady*). By focusing on signals, dynamic $k$ skips tokens with a relatively high score but that can be attributed to the same signal of a neighboring salient token.

## 6  Discussion and Conclusion

Attribution methods disagree on the salience of tokens. Our analyses show that the observed level of agreement is sensitive to the number of $k$ tokens taken into consideration. We propose a dynamic $k$ that can be directly applied to any attribution profile. In contrast to fixed $k$, it takes local relative differences of the attribution values into account. Our analyses with dynamic $k$ indicate that different attribution methods capture varying degrees of attribution scope. Determining dynamic $k$ purely on attribution profiles yields a level of plausibility that is comparable to determining the average human

preference for $k$ and is therefore a viable alternative in the absence of task-specific human data. Furthermore, as dynamic $k$ is estimated for each instance separately, it can account for sentence length bias.

Our peak detection method is an intuitive approach for determining salient tokens based solely on attribution profiles. It focuses on isolated key tokens which might not adequately capture human tendencies of chunking words into phrases. In future work, we plan to analyze how our findings generalize to other task and model conditions and want to explore alternative methods to dynamically determine $k$ by combining the attribution with linguistic information towards better span-level visualisations (see Figure 5). This line of research needs to be closely coupled with cognitive analyses of human preferences.

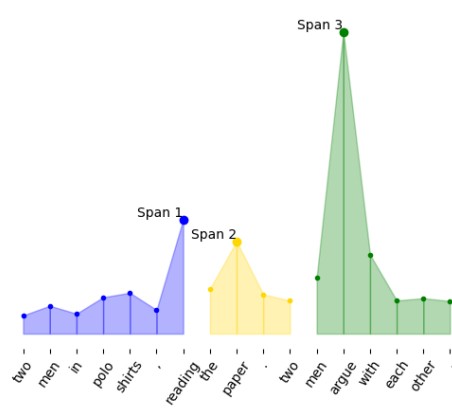

Figure 5: Towards future research on span-based relevance based on dynamic $k$.

## Limitations

In this study, we encountered certain limitations that should be taken into account when interpreting the results and when conducting subsequent research. First, due to pragmatic constraints, we focused on a single model and a selected set of attribution methods for comparison, which restricts the direct generalisation of our findings to methods outside this set. However, the proposed metric agreement@$k$ and the concept of dynamic $k$ can be readily applied to evaluate other methods in future research. The scarce availability of multiple human rationales at the token level, necessary for creating human aggregation scores, limited our ability to expand the scope of this research. Furthermore, it is worth noting that the aggregation scores in our study fall within the range of $[0, .33, .67, 1]$.

Consequently, the precision of the overlap between detected human peaks may be compromised when the number of annotators is low. The resolution of ties in these scores was resolved randomly, which introduces a potential source of improvement and variability in the results. While these limitations should be acknowledged, they do not invalidate the overall contributions of our research. They provide valuable insights into the effectiveness of the selected methods and highlight avenues for future investigations, such as incorporating additional datasets.

## Ethics Statement

In the field of interpretability, results need to be communicated with particular caution to avoid anthropomorphizing neural models. With respect to this study, caution should be exercised when interpreting findings from attribution methods. Attribution scores cannot be blindly relied upon to precisely determine model functioning, as they can be influenced by experimental factors such as task and model performance. To avoid drawing generalised conclusions, it is advisable to employ multiple metrics when studying feature attribution.

## Acknowledgements

Jonathan Kamp's research was funded by the Dutch National Science Organisation (NWO) through the project InDeep: Interpreting Deep Learning Models for Text and Sound (NWA.1292.19.399). Antske Fokkens was supported by the EU Horizon 2020 project InTaVia: In/Tangible European Heritage - Visual Analysis, Curation and Communication (http://intavia.eu) under grant agreement No. 101004825. Lisa Beinborn's work was funded by the Dutch National Science Organisation (NWO) through the VENI program (Vl.Veni.211C.039). We would like to thank the anonymous reviewers for their valuable contribution.

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

## A  Average Pairwise Difference (APD)

The Average Pairwise Difference (APD) indicates how the sets of attribution scores yielded by different combinations of classification models with attribution methods differ from one another.

First, it is possible to compute the Average Difference (AD) between two matrices $T1$ and $T2$ of the same size. AD is a measure of dissimilarity and provides an indication of the overall average magnitude of differences between the matrices. In our case, $T1$ and $T2$ are two attribution matrices and are constructed by concatenating the vectors of token-wise attribution scores $\mathbf{a} = \{a_1, a_2, ..., a_n\}$ (computed for all $l$ attribution methods $A$) for each instance in our dataset of size $d$, after 0-padding $a$s to maximum sentence length. More precisely, let $T1$ and $T2$ be two matrices of size $(d * l) \times m$, where $n$ is the number of sentences in the dataset, $l$ is the number of attribution methods and $m$ is maximum sentence length. The AD between matrices $T1$ and $T2$ is calculated by taking the average of the element-wise absolute differences between the corresponding elements of $T1$ and $T2$.

We then construct two attribution matrices and calculate AD for every pair of runs from the pool of 10 models each trained with a different random seed, assigning an APD score to each model by averaging the AD scores for that model's attribution matrix in pairwise relation to the other models' attribution matrices. The model with lowest APD (in bold in Table 2) was selected for our experiments.

| run_# | DistilBERT |
|-------|------------|
| run_1 | 0.00613 |
| run_2 | 0.00620 |
| run_3 | 0.00611 |
| run_4 | 0.00599 |
| run_5 | 0.00606 |
| run_6 | 0.00628 |
| run_7 | 0.00614 |
| run_8 | **0.00595** |
| run_9 | 0.00604 |
| run_10 | 0.00621 |

Table 2: Average Pairwise Difference between attribution scores produced by different runs trained on 10 different random seeds.

## B  Fine-tuning and Analysis

The input instances for fine-tuning are premises and hypotheses concatenated by a single [SEP] token. We removed 6 instances from the training set where the hypothesis was missing. The main hyperparameters for our models are the following: 15 training epochs with early stopping, training batch size of 32, learning rate set to 5e-6, weight decay set to

0.01 and warmup steps set to 6% of the total. We found that computing the attributions for a larger model such as RoBERTa (Liu et al., 2019) takes significantly longer and aligning the attributions with human annotated text is less straightforward for tokenisation reasons. When pre-processing the human annotations, we assign a 0 score to punctuation characters as they did not receive a dedicated annotation label.

## C    Sentence Length Bias

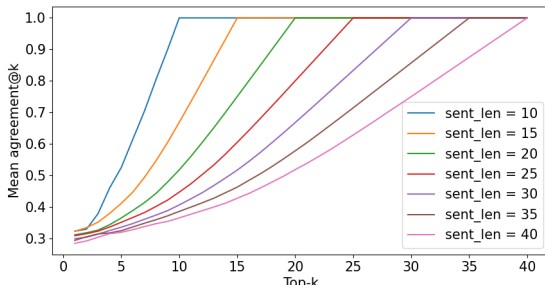

Figure 6: Sentence length bias on overall agreement between methods at different values of top-$k$, for different groups of instances based on sentence length.