# OpenReview forum: "Dynamic Top-k Estimation Consolidates Disagreement between Feature Attribution Methods"
_EMNLP/2023/Conference — EMNLP 2023 Main_

### Official Review · Reviewer_EnSP · 2023-08-04

**Soundness:** 3

**Excitement:**

4: Strong: This paper deepens the understanding of some phenomenon or lowers the barriers to an existing research direction.

**Missing References:**

Related work regarding event/anomaly detection should be included. To provide and example, two such studies are:
* https://arxiv.org/abs/2007.02500
* https://www.nature.com/articles/s41598-021-03526-y

**Paper Topic And Main Contributions:**

This paper proposes an evaluation measure for feature attribution methods along with a way to define the optimum number of attributing features dynamically. The authors used DistilBERT for NLI, on a single dataset, comparing gradient (four: vanilla or integrated, plain or by multiplying with the input embeddings) and perturbation (two: partition SHAP or LIME) methods. The authors proposed using the ratio of attribution methods comprising the token in question, averaged across the sentence, as a measure of relevance/agreement. By studying different attribution methods and human annotations, the authors conclude that (i) vanilla gradient attribution agrees more with the human annotations compared to the rest; (ii) perturbation methods perform better than other gradient methods; (iii) higher agreement is achieved for higher k. The study also implements a basic event detection algorithm to define the number of k automatically.

**Questions For The Authors:**

A. The denominator of eq. 2 is always positive. What have I missed?

B. The mean agreement@k is said to be obtained by averaging over sentence-level agreement@k, but relevance depends on attribution methods by definition (nominator of Eq. 1), so where is this averaging computed exactly?


**Reasons To Accept:**

* In depth study of attribution, contradicting few and verifying other studies in the field.
* Evaluation of attribution in relation with k, a parameter that is overlooked, according to the authors.


**Reasons To Reject:**

* The equation of agreement@k, which is used to compute most of the results in this study, is not clear. Based on the response of the authors, I will need to revisit my assessment regarding this weakness.
* The authors overlook existing work in event/anomaly detection (e.g.: https://arxiv.org/abs/2007.02500 or https://www.nature.com/articles/s41598-021-03526-y) and implement a new algorithm to define the optimum k dynamically.


**Reproducibility:**

4: Could mostly reproduce the results, but there may be some variation because of sample variance or minor variations in their interpretation of the protocol or method.

**Reviewer Confidence:**

3: Pretty sure, but there's a chance I missed something. Although I have a good feel for this area in general, I did not carefully check the paper's details, e.g., the math, experimental design, or novelty.

**Typos Grammar Style And Presentation Improvements:**

* L074: what kind of agreement?
* L090: some intuition about the metric (do you perhaps mean measure?) could be shared
* L098: please define the agreement; human-vs-method or something else?
* L127: please consider renaming model to backbone model, in order to distinguish from attribution models
* L127: the motivation for why DistilBERT was selected (e.g. , instead of BERT) is missing
* Eq.1 & Eq.1: it is relevance that depends on k, not agreement while the latter is rather an average - hence, my understanding is that it is relevance@k and average-relevance (or sentence-relevance).
 * L180: please provide a figure (e.g., in the appendix) where the legend is more informative and each curve has a name.
* L236: please define surrounding; one, two, three tokens?
* Figure 3: this is an interesting figure, but please consider re-producing by using a consistent color scale

---

> ### Author Rebuttal · Authors · 2023-08-27
>
> Message to all reviewers:
>
> We would like to inform the reviewers that we identified an indexing error in our code. This error affects the column and row in Figure 3 that reports human-method agreement. In particular, the values for GradXI and IntGrad now slightly decrease instead of the increase that we reported in the original submission. We will update the figure in the next version and the sentence at lines 247-248 accordingly, i.e. by removing “and with humans”.
>
> This does not alter the contributions listed in the introduction nor the conclusion. If these values were important for your assessment, we understand that you might revisit it.
>
> —
>
> A: We agree with you that we should have been more explicit. If a token has not been included in the top-k selection of any method, it is assigned a relevance score of 0 and it should not be considered in the calculation of agreement@k (hence, $r(w_i)>0$ at the denominator of Eq. 2). The reason for this is that we wanted to avoid artificially inflating the agreement@k score by having high agreement on non-relevant tokens. We will modify Eq. 1 in the next version so that it is explicit that a 0 score is assigned to non-relevant tokens.
>
> B: You are right that the calculation of mean agreement@k, which averages over sentence-level agreement@k scores, is missing in the equations and only appears in-text. With the extra page in the next version, we will add this third equation.
>
> Thank you for the suggestion of including more references to event detection in series. We agree that a more elaborate discussion on event detection would strengthen the paper and we will include studies such as Benkő et al. (2022) and Pang et al. (2021) in the next version. However, the methods outlined in these papers are less suited for event detection in attribution profiles compared to the approach we based our method on, as established by Palshikar et al. (2009). We will explain more clearly in the related work section why that is. In short:
>
> - Benkő et al. (2022) propose a method that finds a unicorn (a single anomaly) that is rare and unique, whereas the type of events described in our study does not meet that description. We are looking for potentially multiple tokens that receive a high signal from attribution methods.
> - The survey by Pang et al. (2021) would help us further define attribution profiles as series; however, DL methods are less suited given the relatively small size of sentences compared to the number of data points in e.g. a stock market series, and given the meaningfulness of the attributions that is only local to the specific sentence.
>
> Finally, thank you for the remaining improvements that you propose. We will take them into account.

---

### Official Review · Reviewer_Cz9e · 2023-08-04

**Soundness:** 3

**Excitement:**

4: Strong: This paper deepens the understanding of some phenomenon or lowers the barriers to an existing research direction.

**Paper Topic And Main Contributions:**

The paper proposes an approach for dynamically determining (sentence-wise) the top k tokens to be displayed on feature attribution methods. It also conducts empirical analysis on the role of the number k on the observed method–method agreement and method–human agreement for extracting explanations for a natural language inference task. The results confirm the hypothesis that the level of agreement between different attribution methods is sensitive to k, and that perturbation-based methods lead to higher agreement than gradient-based ones.

**Reasons To Accept:**

The study seeks to answer a relevant question regarding feature attribution methods applied to language models. Methodology and empirical evaluation are carefully explained, and the conclusions are supported be the empirical results, to the limited scope of the experimental setup. The paper is overall very well written.

**Reasons To Reject:**

There is no specific reason to reject this paper in my opinion.
While not a reason to reject and rightly disclaimed by the authors, experimenting on a single model significantly reduces the scope of the findings, thus limiting the score to be given to this paper.

**Reproducibility:**

4: Could mostly reproduce the results, but there may be some variation because of sample variance or minor variations in their interpretation of the protocol or method.

**Reviewer Confidence:**

3: Pretty sure, but there's a chance I missed something. Although I have a good feel for this area in general, I did not carefully check the paper's details, e.g., the math, experimental design, or novelty.

---

> ### Author Rebuttal · Authors · 2023-08-27
>
> Message to all reviewers:
>
> We would like to inform the reviewers that we identified an indexing error in our code. This error affects the column and row in Figure 3 that reports human-method agreement. In particular, the values for GradXI and IntGrad now slightly decrease instead of the increase that we reported in the original submission. We will update the figure in the next version and the sentence at lines 247-248 accordingly, i.e. by removing “and with humans”.
>
> This does not alter the contributions listed in the introduction nor the conclusion. If these values were important for your assessment, we understand that you might revisit it.
>
> —
>
> Thank you for your review. We agree with you on the fact that including more types of models would expand the scope of the results, and we will leave this exploration to future work.

---

### Official Review · Reviewer_HqLk · 2023-08-11

**Soundness:** 4

**Excitement:**

4: Strong: This paper deepens the understanding of some phenomenon or lowers the barriers to an existing research direction.

**Paper Topic And Main Contributions:**

This paper presents a novel approach, which is motivated by event detection in time series and span-based peak detection, to find salient tokens from attribution profiles across sentences by dynamically selecting the number of optimal k (tokens) from sequential properties. In comparison to previous methods that used a static k, the approach presented in this paper offers several advantages: (1) It avoids sentence-length bias inherent in static k methods; (2) It consolidates the estimation of disagreement among feature attribution methods.

**Reasons To Accept:**

1. The paper is well-written, effectively delineating existing issues and establishing the need for the proposed method.
2. The dynamic k selection approach provides valuable insights into the disparities among attribution methods in terms of token salience.

**Reasons To Reject:**

1. I don't have compelling reasons to oppose this paper at the conference.

**Reproducibility:**

3: Could reproduce the results with some difficulty. The settings of parameters are underspecified or subjectively determined; the training/evaluation data are not widely available.

**Reviewer Confidence:**

2: Willing to defend my evaluation, but it is fairly likely that I missed some details, didn't understand some central points, or can't be sure about the novelty of the work.

---

> ### Author Rebuttal · Authors · 2023-08-27
>
> Message to all reviewers:
>
> We would like to inform the reviewers that we identified an indexing error in our code. This error affects the column and row in Figure 3 that reports human-method agreement. In particular, the values for GradXI and IntGrad now slightly decrease instead of the increase that we reported in the original submission. We will update the figure in the next version and the sentence at lines 247-248 accordingly, i.e. by removing “and with humans”.
>
> This does not alter the contributions listed in the introduction nor the conclusion. If these values were important for your assessment, we understand that you might revisit it.
>
> —
>
> Thank you for your review.

---

### Official Review · Reviewer_Mndj · 2023-08-11

**Soundness:** 4

**Excitement:**

3: Ambivalent: It has merits (e.g., it reports state-of-the-art results, the idea is nice), but there are key weaknesses (e.g., it describes incremental work), and it can significantly benefit from another round of revision. However, I won't object to accepting it if my co-reviewers champion it.

**Paper Topic And Main Contributions:**

This work presents a simple method for optimizing the number of tokens selected to be shown to humans when presenting results from explainability methods that attribute model performance and predictions to specific tokens of the input. They test this approach by using it with Vanilla Gradients, Integrated Gradients, and GradientXInput methods for feature attribution for the SNLI dataset. They find that a dynamically adapted value of k leads to better agreement metrics for both method-method and method-human settings.

**Reasons To Accept:**

The paper presents an interesting analysis of the effect of the number of tokens (k) selected and their selection criteria in improving the agreement for feature attribution using multiple interpretability methods that map model predictions to input features.  The paper is very clearly written and easy to understand, overall.

**Reasons To Reject:**

For experiments with dynamic k, a few more examples from the dataset with the resulting k values from the dynamic selection process as well as corresponding agreement scores could provide further support to the discussion arguments.

**Reproducibility:**

4: Could mostly reproduce the results, but there may be some variation because of sample variance or minor variations in their interpretation of the protocol or method.

**Reviewer Confidence:**

2: Willing to defend my evaluation, but it is fairly likely that I missed some details, didn't understand some central points, or can't be sure about the novelty of the work.

---

> ### Author Rebuttal · Authors · 2023-08-27
>
> Message to all reviewers:
>
> We would like to inform the reviewers that we identified an indexing error in our code. This error affects the column and row in Figure 3 that reports human-method agreement. In particular, the values for GradXI and IntGrad now slightly decrease instead of the increase that we reported in the original submission. We will update the figure in the next version and the sentence at lines 247-248 accordingly, i.e. by removing “and with humans”.
>
> This does not alter the contributions listed in the introduction nor the conclusion. If these values were important for your assessment, we understand that you might revisit it.
>
> —
>
> Yes, we agree with you that extra examples would be a very nice addition. With the additional page of the next version, we will dedicate more space to them. Examples that we would like to discuss:
>
> - Agreement@k with low k (risk of not covering truly relevant tokens) versus high k (risk of sentence length bias) versus dynamic k.
> - Dynamic k not assigning relevance to tokens with negative attribution scores (whereas fixed k does).
> - The selection of tokens by dynamic k that would not make it into a relatively high (e.g. 8) fixed k selection. This relates to the span-based intuition described in Figure 4.

---

### Meta-Review · Area_Chair_ikcu · 2023-09-19

**Recommendation:** 5

**Metareview:**

The authors propose a method which determines the optimal number of tokens (k) which should be displayed to humans based on feature importance attribution scores. The authors perform their experiments on six explainability methods (SHAP, LIME, vanilla-grad, int-grad, the latter two in the plain & gradXinput form) on one model (DistilBERT) and one dataset (eSNLI). The authors show that using their dynamic method for determining k leads to better agreement and avoids the sentence-length bias.

The reviewers agree that the paper is well written, the gap is well motivated and the problem is relevant (Mndj, HqLk, Cz9e). Furthermore, the reviewers commend an interesting analysis which produces valuable insights and answers a relevant question (Mndj, HqLk, EnSP, Cz9e).

There are relatively few criticisms of the paper in the reviews, with two reviewers even noting they cannot find any reason to reject the paper (Cz9e, HqLk) barring perhaps the narrow experimental scope (Cz9e). The main criticisms were omission of some related work (EnSP9), unclarity within an equation (EnSP) and a request for a few more illustrative examples (Mndj). All of these criticisms were addressed in the discussion period, and the authors have either resolved them through comments or committed to include additional material given an extra page.

---

### Decision · Program_Chairs · 2023-10-07

**Decision:**

Accept-Main

**Comment:**

The authors propose a method which determines the optimal number of tokens (k) which should be displayed to humans based on feature importance attribution scores. The authors perform their experiments on six explainability methods (SHAP, LIME, vanilla-grad, int-grad, the latter two in the plain & gradXinput form) on one model (DistilBERT) and one dataset (eSNLI). The authors show that using their dynamic method for determining k leads to better agreement and avoids the sentence-length bias.

The reviewers agree that the paper is well written, the gap is well motivated and the problem is relevant (Mndj, HqLk, Cz9e). Furthermore, the reviewers commend an interesting analysis which produces valuable insights and answers a relevant question (Mndj, HqLk, EnSP, Cz9e).

There are relatively few criticisms of the paper in the reviews, with two reviewers even noting they cannot find any reason to reject the paper (Cz9e, HqLk) barring perhaps the narrow experimental scope (Cz9e). The main criticisms were omission of some related work (EnSP9), unclarity within an equation (EnSP) and a request for a few more illustrative examples (Mndj). All of these criticisms were addressed in the discussion period, and the authors have either resolved them through comments or committed to include additional material given an extra page.